# 3D Object Proposals for Accurate Object Class Detection

**Xiaozhi Chen**[1]    **Kaustav Kundu** [2]    **Yukun Zhu**[2]    **Andrew Berneshawi**[2]

**Huimin Ma**[1]    **Sanja Fidler**[2]    **Raquel Urtasun**[2]

[1]Department of Electronic Engineering
Tsinghua University

[2]Department of Computer Science
University of Toronto

chenxz12@mails.tsinghua.edu.cn, {kkundu, yukun}@cs.toronto.edu,
andrew.berneshawi@mail.utoronto.ca, mhmpub@tsinghua.edu.cn,
{fidler, urtasun}@cs.toronto.edu

## Abstract

The goal of this paper is to generate high-quality 3D object proposals in the context of autonomous driving. Our method exploits stereo imagery to place proposals in the form of 3D bounding boxes. We formulate the problem as minimizing an energy function encoding object size priors, ground plane as well as several depth informed features that reason about free space, point cloud densities and distance to the ground. Our experiments show significant performance gains over existing RGB and RGB-D object proposal methods on the challenging KITTI benchmark. Combined with convolutional neural net (CNN) scoring, our approach outperforms all existing results on all three KITTI object classes.

## 1 Introduction

Due to the development of advanced warning systems, cameras are available onboard of almost every new car produced in the last few years. Computer vision provides a very cost effective solution not only to improve safety, but also to one of the holy grails of AI, fully autonomous self-driving cars. In this paper we are interested in 2D and 3D object detection for autonomous driving.

With the large success of deep learning in the past years, the object detection community shifted from simple appearance scoring on exhaustive sliding windows [1] to more powerful, multi-layer visual representations [2, 3] extracted from a smaller set of object/region proposals [4, 5]. This resulted in over 20% absolute performance gains [6, 7] on the PASCAL VOC benchmark [8].

The motivation behind these bottom-up grouping approaches is to provide a moderate number of *region proposals* among which at least a few accurately cover the ground-truth objects. These approaches typically over-segment an image into super pixels and group them based on several similarity measures [4, 5]. This is the strategy behind Selective Search [4], which is used in most state-of-the-art detectors these days. Contours in the image have also been exploited in order to locate object proposal boxes [9]. Another successful approach is to frame the problem as energy minimization where a parametrized family of energies represents various biases for grouping, thus yielding multiple diverse solutions [10].

Interestingly, the state-of-the-art R-CNN approach [6] does not work well on the autonomous driving benchmark KITTI [11], falling significantly behind the current top performers [12, 13]. This is due to the low achievable recall of the underlying box proposals on this benchmark. KITTI images contain many small objects, severe occlusion, high saturated areas and shadows. Furthermore, KITTI's evaluation requires a much higher overlap with ground-truth for cars in order for a detection to count as correct. Since most existing object/region proposal methods rely on grouping super pixels based on intensity and texture, they fail in these challenging conditions.

| Image | Stereo | depth-Feat | Prior |
|---|---|---|---|

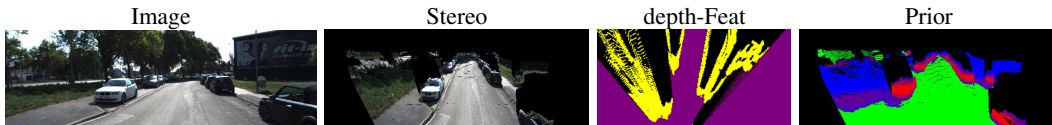

Figure 1: **Features:** From left to right: original image, stereo reconstruction, depth-based features and our prior. In the third image, purple is free space ($F$ in Eq. (2)) and occupancy is yellow ($S$ in Eq. (1)). In the prior, the ground plane is green and red to blue indicates distance to the ground.

In this paper, we propose a new object proposal approach that exploits stereo information as well as contextual models specific to the domain of autonomous driving. Our method reasons in 3D and places proposals in the form of 3D bounding boxes. We exploit object size priors, ground plane, as well as several depth informed features such as free space, point densities inside the box, visibility and distance to the ground. Our experiments show a significant improvement in achievable recall over the state-of-the-art at all overlap thresholds and object occlusion levels, demonstrating that our approach produces highly accurate object proposals. In particular, we achieve a $25\%$ higher recall for 2K proposals than the state-of-the-art RGB-D method MCG-D [14]. Combined with CNN scoring, our method outperforms all published results on object detection for *Car*, *Cyclist* and *Pedestrian* on KITTI [11]. Our code and data are online: `http://www.cs.toronto.edu/~3dop`.

## 2   Related Work

With the wide success of deep networks [2, 3], which typically operate on a fixed spatial scope, there has been increased interest in object proposal generation. Existing approaches range from purely RGB [4, 9, 10, 5, 15, 16], RGB-D [17, 14, 18, 19], to video [20]. In RGB, most approaches combine superpixels into larger regions based on color and texture similarity [4, 5]. These approaches produce around 2,000 proposals per image achieving nearly perfect achievable recall on the PASCAL VOC benchmark [8]. In [10], regions are proposed by defining parametric affinities between pixels and solving the energy using parametric min-cut. The proposed solutions are then scored using simple Gestalt-like features, and typically only 150 top-ranked proposals are needed to succeed in consequent recognition tasks [21, 22, 7]. [16] introduces learning into proposal generation with parametric energies. Exhaustively sampled bounding boxes are scored in [23] using several "objectness" features. BING [15] proposals also score windows based on an object closure measure as a proxy for "objectness". Edgeboxes [9] score millions of windows based on contour information inside and on the boundary of each window. A detailed comparison is done in [24].

Fewer approaches exist that exploit RGB-D. [17, 18] extend CPMC [10] with additional affinities that encourage the proposals to respect occlusion boundaries. [14] extends MCG [5] to 3D by an additional set of depth-informed features. They show significant improvements in performance with respect to past work. In [19], RGB-D videos are used to propose boxes around very accurate point clouds. Relevant to our work is Sliding Shapes [25], which exhaustively evaluates 3D cuboids in RGB-D scenes. This approach, however, utilizes an object scoring function trained on a large number of rendered views of CAD models, and uses complex class-based potentials that make the method run slow in both training and inference. Our work advances over prior work by exploiting the typical sizes of objects in 3D, the ground plane and very efficient depth-informed scoring functions.

Related to our work are also detection approaches for autonomous driving. In [26], objects are pre-detected via a poselet-like approach and a deformable wireframe model is then fit using the image information inside the box. Pepik et al. [27] extend the Deformable Part-based Model [1] to 3D by linking parts across different viewpoints and using a 3D-aware loss function. In [28], an ensemble of models derived from visual and geometrical clusters of object instances is employed. In [13], Selective Search boxes are re-localized using top-down, object level information. [29] proposes a holistic model that re-reasons about DPM detections based on priors from cartographic maps. In KITTI, the best performing method so far is the recently proposed 3DVP [12] which uses the ACF detector [30] and learned occlusion patters in order to improve performance of occluded cars.

## 3   3D Object Proposals

The goal of our approach is to output a diverse set of object proposals in the context of autonomous driving. Since 3D reasoning is of crucial importance in this domain, we place our proposals in 3D and represent them as cuboids. We assume a stereo image pair as input and compute depth via the

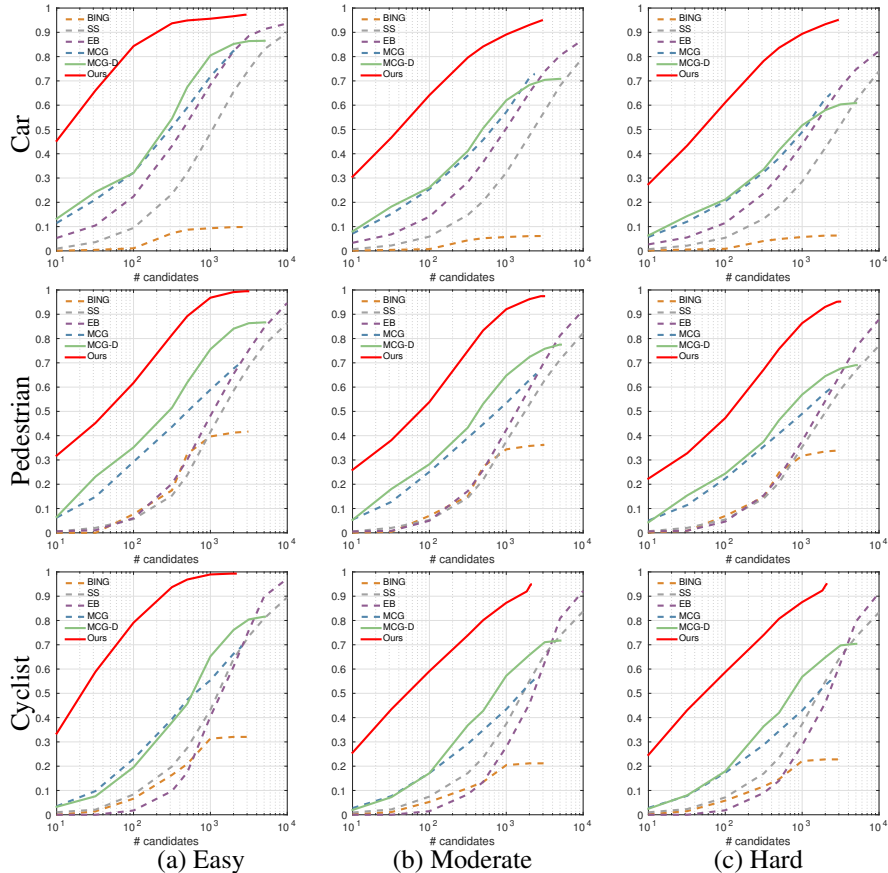

Figure 2: **Proposal recall:** We use 0.7 overlap threshold for *Car* , and 0.5 for *Pedestrian* and *Cyclist*.

state-of-the-art approach by Yamaguchi et al. [31]. We use depth to compute a point-cloud $\mathbf{x}$ and conduct all our reasoning in this domain. We next describe our notation and present our framework.

### 3.1   Proposal Generation as Energy Minimization

We represent each object proposal with a 3D bounding box, denoted by $\mathbf{y}$, which is parametrized by a tuple, $(x, y, z, \theta, c, t)$, where $(x, y, z)$ denotes the center of the 3D box and $\theta$, represents its azimuth angle. Note that each box $\mathbf{y}$ in principle lives in a continuous space, however, for efficiency we reason in a discretized space (details in Sec. 3.2). Here, $c$ denotes the object class of the box and $t \in \{1, \ldots, T_c\}$ indexes the set of 3D box "templates" which represent the physical size variations of each object class $c$. The templates are learned from the training data.

We formulate the proposal generation problem as inference in a Markov Random Field (MRF) which encodes the fact that the proposal $\mathbf{y}$ should enclose a high density region in the point cloud. Furthermore, since the point cloud represents only the visible portion of the 3D space, $\mathbf{y}$ should not overlap with the free space that lies within the rays between the points in the point cloud and the camera. If that was the case, the box would in fact occlude the point cloud, which is not possible. We also encode the fact that the point cloud should not extend vertically beyond our placed 3D box, and that the height of the point cloud in the immediate vicinity of the box should be lower than the box. Our MRF energy thus takes the following form:

$$E(\mathbf{x}, \mathbf{y}) = \mathbf{w}_{c,pcd}^{\top}\phi_{pcd}(\mathbf{x}, \mathbf{y}) + \mathbf{w}_{c,fs}^{\top}\phi_{fs}(\mathbf{x}, \mathbf{y}) + \mathbf{w}_{c,ht}^{\top}\phi_{ht}(\mathbf{x}, \mathbf{y}) + \mathbf{w}_{c,ht-contr}^{\top}\phi_{ht-contr}(\mathbf{x}, \mathbf{y})$$

Note that our energy depends on the object class via class-specific weights $\mathbf{w}_c^{\top}$, which are trained using structured SVM [32] (details in Sec. 3.4). We now explain each potential in more detail.

**Point Cloud Density:**   This potential encodes the density of the point cloud within the box

$$\phi_{pcd}(\mathbf{x}, \mathbf{y}) = \frac{\sum_{p \in \Omega(\mathbf{y})} S(p)}{|\Omega(\mathbf{y})|} \tag{1}$$

where $S(p)$ indicates whether the voxel $p$ is occupied or not (contains point cloud points), and $\Omega(\mathbf{y})$ denotes the set of voxels inside the box defined by $\mathbf{y}$. Fig. 1 visualizes the potential. This potential simply counts the fraction of occupied voxels inside the box. It can be efficiently computed in constant time via integral accumulators, which is a generalization of integral images to 3D.

**Free Space:** This potential encodes the constraint that the free space between the point cloud and the camera cannot be occupied by the box. Let $F$ represent a free space grid, where $F(p) = 1$ means that the ray from the camera to the voxel $p$ does not hit an occupied voxel, i.e., voxel $p$ lies in the free space. We define the potential as follows:

$$\phi_{fs}(\mathbf{x}, \mathbf{y}) = \frac{\sum_{p \in \Omega(\mathbf{y})} (1 - F(p))}{|\Omega(\mathbf{y})|} \tag{2}$$

This potential thus tries to minimize the free space inside the box, and can also be computed efficiently using integral accumulators.

**Height Prior:** This potential encodes the fact that the height of the point cloud inside the box should be close to the mean height of the object class $c$. This is encoded in the following way:

$$\phi_{ht}(\mathbf{x}, \mathbf{y}) = \frac{1}{|\Omega(\mathbf{y})|} \sum_{p \in \Omega(\mathbf{y})} H_c(p) \tag{3}$$

with

$$H_c(p) = \begin{cases} \exp\left[-\frac{1}{2}\left(\frac{d_p - \mu_{c,ht}}{\sigma_{c,ht}}\right)^2\right], & \text{if } S(p) = 1 \\ 0, & \text{o.w.} \end{cases} \tag{4}$$

where, $d_p$ indicates the height of the road plane lying below the voxel $p$. Here, $\mu_{c,ht}, \sigma_{c,ht}$ are the MLE estimates of the mean height and standard deviation by assuming a Gaussian distribution of the data. Integral accumulators can be used to efficiently compute these features.

**Height Contrast:** This potential encodes the fact that the point cloud that surrounds the bounding box should have a lower height than the height of the point cloud inside the box. This is encoded as:

$$\phi_{ht-contr}(\mathbf{x}, \mathbf{y}) = \frac{\phi_{ht}(\mathbf{x}, \mathbf{y})}{\phi_{ht}(\mathbf{x}, \mathbf{y}^+) - \phi_{ht}(\mathbf{x}, \mathbf{y})} \tag{5}$$

where $\mathbf{y}^+$ represents the cuboid obtained by extending $\mathbf{y}$ by 0.6m in the direction of each face.

### 3.2 Discretization and Accumulators

Our point cloud is defined with respect to a left-handed coordinate system, where the positive Z-axis is along the viewing direction of the camera and the Y-axis is along the direction of gravity. We discretize the continuous space such that the width of each voxel is 0.2m in each dimension. We compute the occupancy, free space and height prior grids in this discretized space. Following the idea of integral images, we compute our accumulators in 3D.

### 3.3 Inference

Inference in our model is performed by minimizing the energy defined in Eq. (**??**):

$$\mathbf{y}^* = \operatorname{argmin}_{\mathbf{y}} E(\mathbf{x}, \mathbf{y})$$

Due to the efficient computation of the features using integral accumulators evaluating each configuration $\mathbf{y}$ takes constant time. Still, evaluating exhaustively in the entire grid would be slow. In order to reduce the search space, we carve certain regions of the grid by skipping configurations which do not overlap with the point cloud. We further reduce the search space along the vertical dimension by placing all our bounding boxes on the road plane, $y = y_{road}$. We estimate the road by partitioning the image into super pixels, and train a road classifier using a neural net with several 2D and 3D features. We then use RANSAC on the predicted road pixels to fit the ground plane. Using the ground-plane considerably reduces the search space along the vertical dimension. However since the points are noisy at large distances from the camera, we sample additional proposal boxes at locations farther than 20m from the camera. We sample these boxes at heights $y = y_{road} \pm \sigma_{road}$, where $\sigma_{road}$ is the MLE estimate of the standard deviation by assuming a Gaussian distribution of

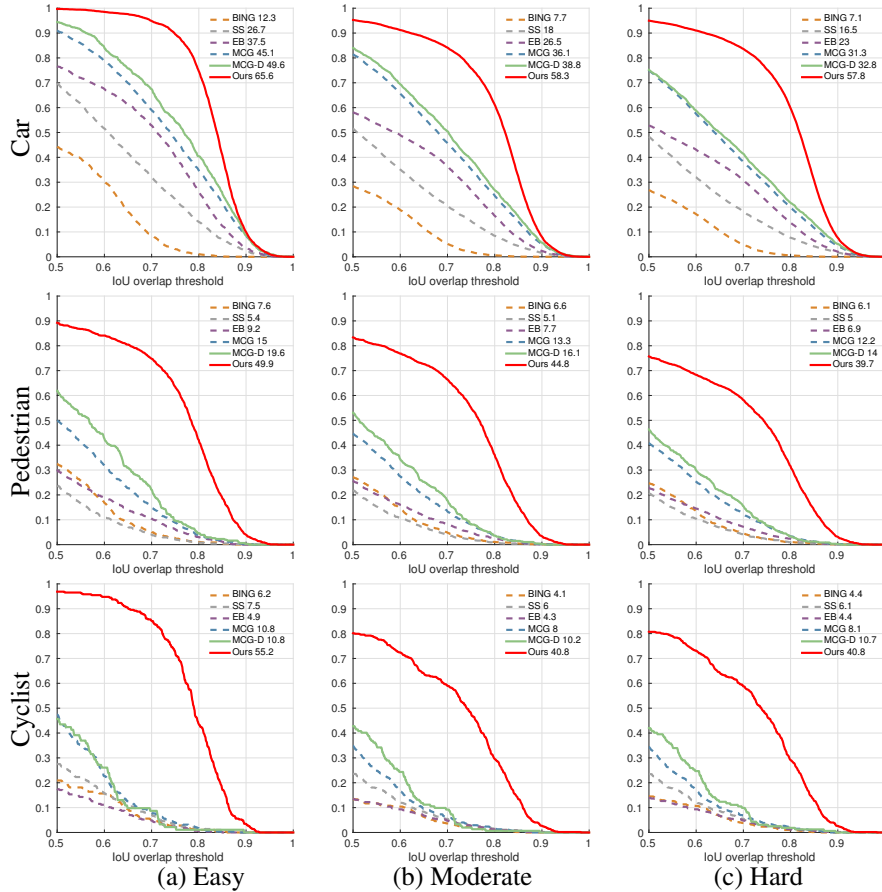

Figure 3: **Recall vs IoU for 500 proposals**. The number next to the labels indicates the average recall (AR).

the distance between objects and the estimated ground plane. Using our sampling strategy, scoring all possible configurations takes only a fraction of a second.

Note that by minimizing our energy we only get *one*, best object candidate. In order to generate $N$ diverse proposals, we sort the values of $E(\mathbf{x}, \mathbf{y})$ for all $\mathbf{y}$, and perform greedy inference: we pick the top scoring proposal, perform NMS, and iterate. The entire inference process and feature computation takes on average 1.2s per image for $N = 2000$ proposals.

### 3.4 Learning

We learn the weights $\{w_{c,pcd}, w_{c,fs}, w_{c,ht}, w_{c,ht-contr}\}$ of the model using structured SVM [32]. Given $N$ ground truth input-output pairs, $\{\mathbf{x}^{(i)}, \mathbf{y}^{(i)}\}_{i=1,\cdots,N}$, the parameters are learnt by solving the following optimization problem:

$$\min_{\mathbf{w} \in \mathbb{R}^D} \frac{1}{2}||\mathbf{w}||^2 + \frac{C}{N}\sum_{i=1}^{N}\xi_i$$

$$\text{s.t.:} \quad \mathbf{w}^T(\phi(\mathbf{x}^{(i)}, \mathbf{y}) - \phi(\mathbf{x}^{(i)}, \mathbf{y}^{(i)})) \geq \Delta(\mathbf{y}^{(i)}, \mathbf{y}) - \xi_i, \quad \forall \mathbf{y} \setminus \mathbf{y}^{(i)}$$

We use the parallel cutting plane of [33] to solve this minimization problem. We use Intersection-over-Union (IoU) between the set of GT boxes, $\mathbf{y}^{(i)}$, and candidates $\mathbf{y}$ as the task loss $\Delta(\mathbf{y}^{(i)}, \mathbf{y})$. We compute IoU in 3D as the volume of intersection of two 3D boxes divided by the volume of their union. This is a very strict measure that encourages accurate 3D placement of the proposals.

### 3.5 Object Detection and Orientation Estimation Network

We use our object proposal method for the task of object detection and orientation estimation. We score bounding box proposals using CNN. Our network is built on Fast R-CNN [34], which share convolutional features across all proposals and use ROI pooling layer to compute proposal-specific

| | Cars | | | Pedestrians | | | Cyclists | | |
|---|---|---|---|---|---|---|---|---|---|
| | Easy | Moderate | Hard | Easy | Moderate | Hard | Easy | Moderate | Hard |
| LSVM-MDPM-sv [35, 1] | 68.02 | 56.48 | 44.18 | 47.74 | 39.36 | 35.95 | 35.04 | 27.50 | 26.21 |
| SquaresICF [36] | - | - | - | 57.33 | 44.42 | 40.08 | - | - | - |
| DPM-C8B1 [37] | 74.33 | 60.99 | 47.16 | 38.96 | 29.03 | 25.61 | 43.49 | 29.04 | 26.20 |
| MDPM-un-BB [1] | 71.19 | 62.16 | 48.43 | - | - | - | - | - | - |
| DPM-VOC+VP [27] | 74.95 | 64.71 | 48.76 | 59.48 | 44.86 | 40.37 | 42.43 | 31.08 | 28.23 |
| OC-DPM [38] | 74.94 | 65.95 | 53.86 | - | - | - | - | - | - |
| AOG [39] | 84.36 | 71.88 | 59.27 | - | - | - | - | - | - |
| SubCat [28] | 84.14 | 75.46 | 59.71 | 54.67 | 42.34 | 37.95 | - | - | - |
| DA-DPM [40] | - | - | - | 56.36 | 45.51 | 41.08 | - | - | - |
| Fusion-DPM [41] | - | - | - | 59.51 | 46.67 | 42.05 | - | - | - |
| R-CNN [42] | - | - | - | 61.61 | 50.13 | 44.79 | - | - | - |
| FilteredICF [43] | - | - | - | 61.14 | 53.98 | 49.29 | - | - | - |
| pAUCEnsT [44] | - | - | - | 65.26 | 54.49 | 48.60 | 51.62 | 38.03 | 33.38 |
| MV-RGBD-RF [45] | - | - | - | 70.21 | 54.56 | 51.25 | 54.02 | 39.72 | 34.82 |
| 3DVP [12] | 87.46 | 75.77 | 65.38 | - | - | - | - | - | - |
| Regionlets [13] | 84.75 | 76.45 | 59.70 | 73.14 | 61.15 | 55.21 | 70.41 | 58.72 | 51.83 |
| Ours | **93.04** | **88.64** | **79.10** | **81.78** | **67.47** | **64.70** | **78.39** | **68.94** | **61.37** |

Table 1: Average Precision (AP) (in %) on the test set of the KITTI Object Detection Benchmark.

| | Cars | | | Pedestrians | | | Cyclists | | |
|---|---|---|---|---|---|---|---|---|---|
| | Easy | Moderate | Hard | Easy | Moderate | Hard | Easy | Moderate | Hard |
| AOG [39] | 43.81 | 38.21 | 31.53 | - | - | - | - | - | - |
| DPM-C8B1 [37] | 59.51 | 50.32 | 39.22 | 31.08 | 23.37 | 20.72 | 27.25 | 19.25 | 17.95 |
| LSVM-MDPM-sv [35, 1] | 67.27 | 55.77 | 43.59 | 43.58 | 35.49 | 32.42 | 27.54 | 22.07 | 21.45 |
| DPM-VOC+VP [27] | 72.28 | 61.84 | 46.54 | 53.55 | 39.83 | 35.73 | 30.52 | 23.17 | 21.58 |
| OC-DPM [38] | 73.50 | 64.42 | 52.40 | - | - | - | - | - | - |
| SubCat [28] | 83.41 | 74.42 | 58.83 | 44.32 | 34.18 | 30.76 | - | - | - |
| 3DVP [12] | 86.92 | 74.59 | 64.11 | - | - | - | - | - | - |
| Ours | **91.44** | **86.10** | **76.52** | **72.94** | **59.80** | **57.03** | **70.13** | **58.68** | **52.35** |

Table 2: AOS scores (in %) on the test set of KITTI's Object Detection and Orientation Estimation Benchmark.

features. We extend this basic network by adding a context branch after the last convolutional layer, and an orientation regression loss to jointly learn object location and orientation. Features output from the original and the context branches are concatenated and fed to the prediction layers. The context regions are obtained by enlarging the candidate boxes by a factor of 1.5. We used smooth $L_1$ loss [34] for orientation regression. We use OxfordNet [3] trained on ImageNet to initialize the weights of convolutional layers and the branch for candidate boxes. The parameters of the context branch are initialized by copying the weights from the original branch. We then fine-tune it end to end on the KITTI training set.

## 4 Experimental Evaluation

We evaluate our approach on the challenging KITTI autonomous driving dataset [11], which contains three object classes: *Car*, *Pedestrian*, and *Cyclist*. KITTI's object detection benchmark has 7,481 training and 7,518 test images. Evaluation is done in three regimes: *easy*, *moderate* and *hard*, containing objects at different occlusion and truncation levels. The moderate regime is used to rank the competing methods in the benchmark. Since the test ground-truth labels are not available, we split the KITTI training set into train and validation sets (each containing half of the images). We ensure that our training and validation set do not come from the same video sequences, and evaluate the performance of our bounding box proposals on the validation set.

Following [4, 24], we use the oracle recall as metric. For each ground-truth (GT) object we find the proposal that overlaps the most in IoU (i.e., "best proposal"). We say that a GT instance has been recalled if IoU exceeds 70% for cars, and 50% for pedestrians and cyclists. This follows the standard KITTI's setup. Oracle recall thus computes the percentage of recalled GT objects, and thus the best achievable recall. We also show how different number of generated proposals affect recall.

**Comparison to the State-of-the-art:** We compare our approach to several baselines: MCG-D [14], MCG [5], Selective Search (SS) [4], BING [15], and Edge Boxes (EB) [9]. Fig. 2 shows recall as a function of the number of candidates. We can see that by using 1000 proposals, we achieve around 90% recall for Cars in the *moderate* and *hard* regimes, while for *easy* we need

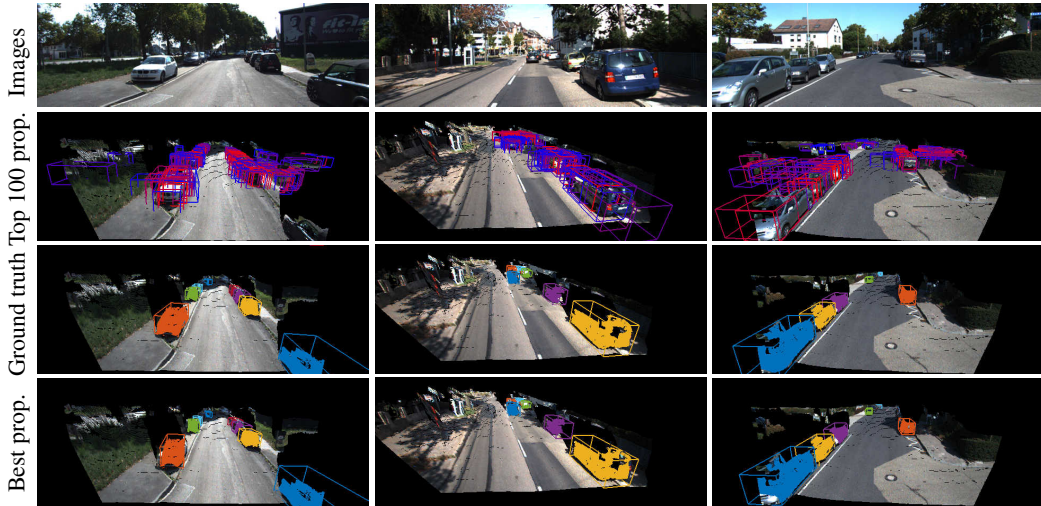

Figure 4: Qualitative results for the Car class. We show the original image, 100 top scoring proposals, ground-truth 3D boxes, and our best set of proposals that cover the ground-truth.

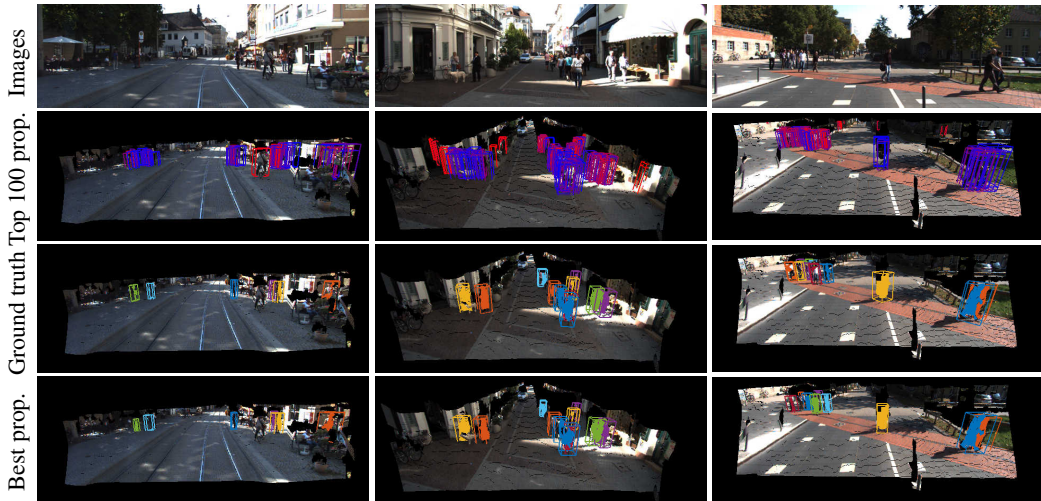

Figure 5: Qualitative examples for the Pedestrian class.

| Method | BING | Selective Search | Edge Boxes (EB) | MCG | MCG-D | **Ours** |
|---|---|---|---|---|---|---|
| Time (seconds) | 0.01 | 15 | 1.5 | 100 | 160 | 1.2 |

Table 3: Running time of different proposal methods

only 200 candidates to get the same recall. Notice that other methods saturate or require orders of magnitude more candidates to reach 90% recall. For Pedestrians and Cyclists our results show similar improvements over the baselines. Note that while we use depth-based features, MCG-D uses both depth and appearance based features, and all other methods use only appearance features. This shows the importance of 3D information in the autonomous driving scenario. Furthermore, the other methods use class agnostic proposals to generate the candidates, whereas we generate them based on the object class. This allows us to achieve higher recall values by exploiting size priors tailored to each class. Fig. 3 shows recall for 500 proposals as a function of the IoU overlap. Our approach significantly outperforms the baselines, particularly for Cyclists.

**Running Time:**    Table 3 shows running time of different proposal methods. Our approach is fairly efficient and can compute all features and proposals in 1.2s on a single core.

**Qualitative Results:**    Figs. 4 and  5 show qualitative results for cars and pedestrians. We show the input RGB image, top 100 proposals, the GT boxes in 3D, as well as proposals from our method with the best 3D IoU (chosen among 2000 proposals). Our method produces very precise proposals even for the more difficult (far away or occluded) objects.

**Object Detection:** To evaluate our full object detection pipeline, we report results on the *test set* of the KITTI benchmark. The results are presented in Table 1. Our approach outperforms all the competitors significantly across all categories. In particular, we achieve 12.19%, 6.32% and 10.22% improvement in AP for *Cars*, *Pedestrians*, and *Cyclists*, in the moderate setting.

**Object Orientation Estimation:** Average Orientation Similarity (AOS) [11] is used as the evaluation metric in object detection and orientation estimation task. Results on KITTI *test set* are shown in Table 2. Our approach again outperforms all approaches by a large margin. Particularly, our approach achieves ~12% higher scores than 3DVP [12] on *Cars* in moderate and hard data. The improvement on *Pedestrians* and *Cyclists* are even more significant as they are more than 20% higher than the second best method.

**Suppl. material:** We refer the reader to supplementary material for many additional results.

## 5 Conclusion

We have presented a novel approach to object proposal generation in the context of autonomous driving. In contrast to most existing work, we take advantage of stereo imagery and reason directly in 3D. We formulate the problem as inference in a Markov random field encoding object size priors, ground plane and a variety of depth informed features. Our approach significantly outperforms existing state-of-the-art object proposal methods on the challenging KITTI benchmark. In particular, for 2K proposals our approach achieves a 25% higher recall than the state-of-the-art RGB-D method MCG-D [14]. Combined with CNN scoring our method significantly outperforms all previous published object detection results for all three object classes on the KITTI [11] benchmark.

**Acknowledgements:** The work was partially supported by NSFC 61171113, NSERC and Toyota Motor Corporation.

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
