[Supplementary Material · top.pdf]

# Supplementary Material: 3D Object Proposals for Accurate Object Class Detection

**Xiaozhi Chen**[*,1]     **Kaustav Kundu** [*,2]     **Yukun Zhu**[2]     **Andrew Berneshawi**[2]

**Huimin Ma**[1]     **Sanja Fidler**[2]     **Raquel Urtasun**[2]

[1]Department of Electronic Engineering
Tsinghua University

[2]Department of Computer Science
University of Toronto

chenxz12@mails.tsinghua.edu.cn, {kkundu, yukun}@cs.toronto.edu,
andrew.berneshawi@mail.utoronto.ca, mhmpub@tsinghua.edu.cn,
{fidler, urtasun}@cs.toronto.edu

In the Supplementary Material we first provide a description of the architecture of our object detection network. We then describe some details about our road estimation method. We also introduce the class-agnostic variant of our approach. We provide comprehensive results on object detection and orientation estimation, and recall statistics for both 2D and 3D box proposals. Finally, a visualization of success cases as well as failure modes are presented, followed by a detailed error analysis of our detector.

## 1   Object Detection and Orientation Estimation Network

The architecture of our proposal scoring CNN for object detection is shown in 1. We built this network using the Fast R-CNN [1] implementation. The network has two branches after the last convolutional layer, one for the candidate boxes and another for the contextual boxes. Both branches are stacked by a ROI pooling layer and two fully connected layers. The context boxes are computed by enlarging the candidate boxes by a factor of 1.5, following the segDeepM approach [2]. We concatenate the features from $fc_7$ layers and feed them to three prediction layers. We learn the category confidences, bounding box location offsets, and object orientation jointly using three task losses. The loss for box classification and regression are the same as used by Fast R-CNN [1]. Smooth $L_1$ loss [1] is used for orientation regression.

We choose the training samples of our network based on the IoU overlap threshold of the 2D proposal bounding boxes and ground0truth boxes. Since KITTI uses different overlap criteria for *Car* and *Pedestrian*/*Cyclist*, we set the threshold for *Car* to 0.7 and 0.5 for the *Pedestrian*/*Cyclist* classes. Because the objects in KITTI images are typically small, we upscale the input image by a factor of 3.5, which has proven to be crucial to achieve very good performance. We use a single scale for input images in both training and testing. At test time, it takes around 2s to evaluate one image with ~2K proposals on a Titan X GPU.

## 2   Road Estimation

We estimated the ground-plane by classifying superpixels using a neural net, and fitting a plane to the estimated ground pixels using RANSAC. We used the following features on the superpixels as input to the network: mean RGB values, average 2D and 3D position (computed via depth), pitch and roll angles relative to the camera of the plane fit to the superpixel, a flag as to whether the average 2D position was above the horizon line, and standard deviation of both the color values and 3D position.

---

[*] Denotes equal contribution

Figure 1: CNN architecture used to score our proposals for object detection.

| Metric | ort | ctx | cls | Cars | | | Pedestrians | | | Cyclists | | |
|---|---|---|---|---|---|---|---|---|---|---|---|---|
| | | | | Easy | Moderate | Hard | Easy | Moderate | Hard | Easy | Moderate | Hard |
| AP | | | ✓ | 92.18 | 87.26 | 78.58 | 72.56 | 69.08 | 61.34 | **90.69** | 62.82 | 58.26 |
| | ✓ | | ✓ | 92.67 | 87.52 | 78.78 | 72.42 | **69.42** | 61.55 | 85.92 | 62.54 | 57.71 |
| | ✓ | ✓ | | 92.76 | 87.30 | 78.61 | **73.76** | 66.26 | **63.15** | 85.91 | 62.82 | 57.05 |
| | ✓ | ✓ | ✓ | **93.08** | **88.07** | **79.39** | 71.40 | 64.46 | 60.39 | 83.82 | **63.47** | **60.93** |
| AOS | | | ✓ | 39.52 | 38.24 | 34.01 | 34.15 | 33.08 | 29.27 | 63.88 | 43.85 | 40.36 |
| | ✓ | | ✓ | 91.46 | **85.80** | 76.73 | **62.25** | **59.15** | **52.24** | **77.60** | **55.75** | 51.23 |
| | ✓ | ✓ | | 91.22 | 85.12 | 75.74 | 61.62 | 55.01 | 52.14 | 74.28 | 53.96 | 49.05 |
| | ✓ | ✓ | ✓ | **91.58** | **85.80** | **76.80** | 61.57 | 54.79 | 51.12 | 73.94 | 55.59 | **53.00** |

Table 1: **Object detection and orientation estimation results on validation set of KITTI**. ort: orientation regression loss; ctx: contextual information; cls: class-specific weights in proposal generation.

## 3 Class-Agnostic Proposals

To study the effect of class-specific weights in proposals generation, we also generate class-agnostic proposals by learning only a single MRF model for all categories. We also learn object templates for all classes jointly rather than for each specific class. Therefore, the weights in this energy are class-independent (we have only a single set of weights). We refer the reader to Sec. 4 and Sec. 5 for a detailed comparison of our two approaches.

## 4 Object Detection and Orientation Estimation Performance

Table 1 shows an ablation study of the effects of using the loss function for orientation, contextual features, and class information in proposal generation. Note that by jointly performing bounding box regression and orientation estimation we get a large boost in performance in terms of the orientation estimation task, while the 2D detection AP remains very similar. When using both orientation loss and contextual features, we get the best AP and AOS for *Cars*. In particular, when removing class information in proposal generation, the detection performance only drops by less than 1% in most cases.

Fig. 2 depicts detection AP and orientation estimation scores on the KITTI **test set**. We compare our method to the state-of-the-art approaches listed on the KITTI detection leaderboard [1].

Figure 2: **Precision-Recall curves on KITTI test set**. The number next to the label indicates the Average Precision (AP).

## 5 Proposal Statistics

In this section, we show a detailed analysis of our object proposals by category and difficulty level (reflected by size and occlusion/truncation levels). We report recall as a function of the number of proposals for 2D bounding boxes in Fig. 4, average recall (AR) as a function of the number of proposals in Fig. 5, recall as a function of the overlap for three different proposal budgets: 500, 1000 and 2000 in Fig. 6, Fig. 7 and Fig. 8, respectively.

We also plot the recall as a function of the distance from the camera in Fig. 9. Notably, our approach maintains a very high recall when the distance from the camera is increasing.

As our approach outputs 3D bounding boxes, we also evaluate recall of the 3D boxes. We plot recall versus number of proposals and recall versus distance from the camera in Fig. 10 and Fig. 11, respectively. We set the IoU threshold to 0.25 when computing 3D overlap between 3D box proposals and 3D ground truth. Note that none of the baselines output 3D bounding boxes, thus we cannot compare our method to any of the available baselines.

**Car (Easy)** — LSVM-MDPM-sv 67.27, DPM-C8B1 59.51, DPM-VOC+VP 72.28, OC-DPM 73.50, AOG 43.81, SubCat 83.41, 3DVP 86.92, Ours 91.44

**Car (Moderate)** — LSVM-MDPM-sv 55.77, DPM-C8B1 50.32, DPM-VOC+VP 61.84, OC-DPM 64.42, AOG 38.21, SubCat 74.42, 3DVP 74.59, Ours 86.10

**Car (Hard)** — LSVM-MDPM-sv 43.59, DPM-C8B1 39.22, DPM-VOC+VP 46.54, OC-DPM 52.40, AOG 31.53, SubCat 58.83, 3DVP 64.11, Ours 76.52

**Pedestrian (Easy)** — DPM-C8B1 31.08, LSVM-MDPM-sv 43.58, DPM-VOC+VP 53.55, Ours 72.94

**Pedestrian (Moderate)** — DPM-C8B1 23.37, LSVM-MDPM-sv 35.49, DPM-VOC+VP 39.83, Ours 59.80

**Pedestrian (Hard)** — DPM-C8B1 20.72, LSVM-MDPM-sv 32.42, DPM-VOC+VP 35.73, Ours 57.03

**Cyclist (Easy)** — LSVM-MDPM-sv 27.54, DPM-C8B1 27.25, DPM-VOC+VP 30.52, Ours 70.13

**Cyclist (Moderate)** — LSVM-MDPM-sv 22.07, DPM-C8B1 19.25, DPM-VOC+VP 23.17, Ours 58.68

**Cyclist (Hard)** — LSVM-MDPM-sv 21.45, DPM-C8B1 17.95, DPM-VOC+VP 21.58, Ours 52.35

Figure 3: **Orientation Similarity-Recall curves on KITTI test set**. The number next to the label indicates the Average Orientation Similarity (AOS).

# 6 Visualization

From Fig. 12 to 15, we show success cases and failure modes.

Figure 4: **2D bounding box Recall vs #Candidates**. We use an overlap threshold of 0.7 for Car, and 0.5 for Pedestrian and Cyclist. From left to right are for easy, moderate, and hard objects, respectively. We also compare with the class-agnostic proposals of our approach, indicated as "Ours⋆".

Figure 5: **Average Recall (AR) vs #Candidates for 2D bounding boxes**. Class-agnostic variant of our approach is indicated as "Ours*".

Figure 6: **2D bounding box Recall vs IoU for 500 proposals**. The number next to the label indicates the average recall (AR). We also compare with the class-agnostic proposals of our approach, indicated as "Ours*".

Figure 7: **2D bounding box Recall vs IoU for 1000 proposals**. The number next to the label indicates the average recall (AR). We also compare with the class-agnostic proposals of our approach, indicated as "Ours*".

Figure 8: **2D bounding box Recall vs IoU for 2000 proposals**. The number next to the label indicates the average recall (AR). We also compare with the class-agnostic proposals of our approach, indicated as "Ours*".

Figure 9: **2D bounding box Recall vs Distance using 2000 proposals**. We use an overlap threshold of 0.7 for Car, and 0.5 for Pedestrian and Cyclist. We also compare with the class-agnostic proposals of our approach, indicated as "Ours*".

Figure 10: **3D bounding box Recall vs #Candidates at IoU threshold of 0.25**. The class-agnostic variant of our approach is indicated as "Ours*".

Figure 11: **3D bounding box Recall vs Distance using 2000 proposals**. We use an overlap threshold of 0.25 for all classes. The class-agnostic variant of our approach is indicated as "Ours⋆".

## 6.1 Success Cases

Figure 12: Qualitative results for the Car class. We show the original image, prior map, 100 top scoring proposals, ground-truth 3D boxes, and our best set of proposals that cover the ground-truth.

Figure 13: Qualitative examples for the Pedestrian class.

Figure 14: Qualitative examples for the Cyclist class.

## 6.2 Failure Cases

(a) Car

(b) Pedestrian

(c) Cyclist

Figure 15: **Failure modes**. For the top row in each category, ground truth boxes and detected boxes are indicated in blue and green, respectively. For the bottom row, correct detection are indicated in blue. The missed objects are indicated in red. Most failure cases are due to stereo estimation errors, especially far from the camera.

## 7 Error Analysis

In this section we show a detailed error analysis of our detector using the methodology of [3].

### 7.1 False Positives

Fig. 16 shows the distribution of top-ranked false positive types. Pedestrian and Cyclist are considered as similar categories. Some top false positive examples are shown in Fig. 17. It indicates that most false positives come from localization errors, especially for Car. This is not surprising as the IOU threshold is very high (i.e., 70%). Pedestrian and Cyclist also easily confuse each other.

Figure 16: **False positive (FP) trends with rank.** Each plot shows fraction of FP of each type as the total number of FP increase. The labels indicate the type of error. "Loc": poor localization ($0.1 \le$ overlap$<0.7$ for Car and $0.1 \le$ overlap$<0.5$ for Pedestrian/Cyclist or duplicate detections); "Sim": confusion with a similar category; "Oth": confusion with a dissimilar object category; "BG": background.

## 7.2 Analysis of Characteristics

We further show a detailed analysis of our detector's performance as a function of different object characteristics in Fig. 18. Using annotations from KITTI, we analyze four characteristics: occlusion, truncation, bounding box area and aspect ratio. Occlusion characteristic is described by three levels using the KITTI annotations, i.e., "N"=no occlusion; "L"=partly occluded; "H"=difficult to see. As KITTI only evaluates objects truncated by less than 50%, we divided truncation into four levels according to the percentage of truncation, i.e., "N"=no truncation; "L"=(0%, 15%]; "M"=(15%, 30%]; "H"=(30%, 50%]. For bounding box area and aspect ratio, we follow the measure proposed by [3]. Fig. 18 and Fig. 19 show that our detectors can be further improved by handling heavily occluded objects as well as small objects.

## Footnotes

[1] http://www.cvlibs.net/datasets/kitti/eval_object.php

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

Car (loc): ov=0.66 1-r=0.99    Car (loc): ov=0.68 1-r=0.97    Car (loc): ov=0.64 1-r=0.90

Car (loc): ov=0.53 1-r=0.90    Car (loc): ov=0.70 1-r=0.90    Car (loc): ov=0.28 1-r=0.90

Car (loc): ov=0.68 1-r=0.90    Car (loc): ov=0.67 1-r=0.89    Car (loc): ov=0.70 1-r=0.88

(a) Car

Pedestrian (loc): ov=0.46 1-r=0.98    Pedestrian (loc): ov=0.12 1-r=0.96    Pedestrian (loc): ov=0.12 1-r=0.91

Pedestrian (loc): ov=0.48 1-r=0.91    Pedestrian (bg): ov=0.00 1-r=0.89    Pedestrian (loc): ov=0.47 1-r=0.84

Pedestrian (loc): ov=0.41 1-r=0.83    Pedestrian (bg): ov=0.05 1-r=0.82    Pedestrian (loc): ov=0.46 1-r=0.82

(b) Pedestrian

Cyclist (loc): ov=0.40 1-r=0.80    Cyclist (sim): ov=0.00 1-r=0.68    Cyclist (sim): ov=0.00 1-r=0.63

Cyclist (loc): ov=0.47 1-r=0.63    Cyclist (sim): ov=0.00 1-r=0.63    Cyclist (sim): ov=0.00 1-r=0.62

Cyclist (bg): ov=0.00 1-r=0.62    Cyclist (loc): ov=0.12 1-r=0.62    Cyclist (bg): ov=0.00 1-r=0.60

(c) Cyclist

Figure 17: **Examples of top false positives.** We show the top 9 false positives (FPs) for each category. The text indicates the type of error. The amount of overlap ("ov') with a true object, and the fraction of correct examples that are ranked lower than the given false positive ("1-r", for 1-recall). Localization errors could be insufficient overlap (less than 0.7 for Car and 0.5 for Pedestrian/Cyclist) or duplicate detections.

Figure 18: **Analysis of characteristics for each category**: We show $AP_N$ (+) with standard error bars (red) for four characteristics: occlusion, truncation, bounding box area, aspect ratio. Black dashed lines indicate overall $AP_N$ .

Figure 19: **Summary of sensitivity and impact of object characteristics**: We show the average (over categories) $AP_N$ performance of the highest performing and lowest performing subsets within each characteristic. Overall $AP_N$ is indicated by the black dashed line. The difference between max and min indicates sensitivity. The difference between max and overall indicates the impact.