[Reviews · NeurIPS 2015]

Submitted by Assigned_Reviewer_1

-- Summary --

This paper addresses the problem of generating 3D object proposals given a stereo image pair from an autonomous driving vehicle. The paper proposes a set of features for a 3D cuboid over a point cloud and ground plane derived from the stereo image pair. The features include point cloud density, free space, object height prior, and object height relative to its surroundings. Note that the features are dependant on knowledge of the object class (other "objectness" proposal methods are agnostic to the object class). A structural SVM is trained to predict the "objectness" of the 3D cuboid proposal. Bounding box and orientation regression is performed for the final object detection.

The approach is evaluated on the KITTI benchmark, and the approach outperforms competing baselines for oracle recall and object detection.

-- Comments --

Positives: Generating good object proposals has shown to be an important step for object detection.

While most prior work has focused on 2D still images, this paper focuses on 3D data, which has arguably received less attention. The approach outperforms baselines in this space (including approaches informed by 3D), which also translates to improved performance for object detection.

The paper is reasonably clear to get the ideas of the approach and cites well prior art in this space.

Negatives: The main ideas behind the paper may be perceived as incremental from a machine learning perspective.

I think the boost in performance would be appreciated more by the 3D vision community.

Detailed comments:

+ Section 3.4: I'm not clear why a structured SVM is used here. Section 3.1 introduced a feature vector, and it seems that only a linear classifier is needed.

I'd like to see how performance of a linear SVM with hard negative mining would perform.

+ Prior work in generating object proposals are object class agnostic.

I'd like to see performance where object class information is not used (e.g. remove object height from feature vector).

+ Line 107: It would be good to clarify that the input is a calibrated stereo image pair since the approach relies on metric information. Also perhaps mention the dependency on ground plane estimation.

+ Paragraph starting Line 142: What is the parameterization of t? The indices for t and \theta are not needed.

Finally, it may be good to mention that it is assumed the boxes live on the ground and/or its base is parallel to the ground plane.

+ Equation (1): It may be good to make explicit that each \phi depends on x.

+ Equation (4): Why not "max" over the heights?

+ Line 194: I think this potential assumes objects are well segmented in 3D (i.e. no overlap or intersection).

Is this a valid assumption?

+ Line 246: Please provide more details on the ground plane estimation, as it is not reproducible as described.

+ Line 255: NMS => non maximum suppression

+ Section 3.4: What is the setting of C?

Also, is "y" supposed to be bold?
Summary: My rating is based on the boost in oracle recall for "car" and "cyclist" object classes on the KITTI benchmark.

Submitted by Assigned_Reviewer_2

This paper presents a new object proposal method that exploits depth data (from stereo) to propose bounding boxes for object (pedestrian, car, and cyclist) detection in self driving car applications. It gets point cloud from stereo pairs and defines a scoring function based on some features that operate on the point cloud. The weights in scoring function are object dependent and are learned in a structured SVM framework. In inference, it evaluates the scoring function for all possible windows that match the prior (being on the road). Finally, an RCNN scores the proposed bounding boxes to detect the objects.

The method is evaluated on KITTI dataset for three objects (pedestrian, car, and cyclist) and have got good results compared to reasonable baselines.

This is an interesting idea and the computer vision community is interested in improving object detection in self driving car applications, so should be interesting to the audience.

The results are great.

Usually, object proposal algorithms are category independent and then later, an object detection algorithm detects the objects by evaluating all the proposals. However, here, the weights are object dependent, so I think it's better to call this paper an object detection algorithm that has a high recall rather than an object proposal one. Since the scoring function doesn't sort the detected boxes well, an RCNN is resorting the detections to get a better average precision. Hence, it might be good to clarify this in the paper. Therefore, there are quite a few object detection algorithms that operate in 3D and can be used as baselines here. For instance [24] uses RGBD to detect objects.

The results are great, but the technical novelty is somewhat limited as it introduces some depth based features, does exhaustive search in inference, and learns using structured SVM. This is ok, but I think it would be better to evaluate this method at least on one more dataset to show that the features can generalize across datasets.

I think good results of this paper is the main positive point.
Summary: This paper shows great results for pedestrian, car, and cyclist detection in KITTI dataset. However, the technical novelty seems to be limited.

Submitted by Assigned_Reviewer_3

The problem of 3D object proposal is much

less well studied than 2D object proposal, and this paper has done a good job on developing effective approach for this task which fundamentally

differ from the popular 2D approach (grouping superpixels, e.g. Selective Search.)

For evaluation, the paper did a good job on comparing with the state of the art approaches. But one missing component would be that although the paper is doing 3D object proposal there is no evaluation done in in 3D (all the detection are evaluate with 2D boxes).

Although the paper focuses on exploiting contextual models specific to autonomous driving, it will be very interesting to see whether this approach can be generalized to other domains (e.g. indoor RGB-D scan). The task is fundamentally similar and the information that the paper makes use of such as object size priors, ground plane, free space, point cloud densities and distance to the ground will still be very useful in general RGB-D indoor scene. The challenges might be the indoor scene will be more clutter and more object categories. It will also make the compare with MCG-D [14] more fair, because it is designed for

indoor RGB-D scan, and haven't apply such strong

domain specific contextual model. Therefore, applying the proposed approach to RGB-D indoor scene will definitely make the paper much stronger.
Summary: The paper develop a novel and effective approach for 3D object proposal which fundamentally

differ from the popular 2D approach e.g. Selective Search. I suggest the paper to include 3D evaluation into the paper, and also show the generalization ability of the approach by applying it into indoor RGB-D scan.

Submitted by Assigned_Reviewer_4

3D object proposal for automatic driving is a novel and less studied research direction. The proposed method is more accurate than other methods. However, 1.2s process time is not particular fast. The paper is in general well-written, but I still have a few questions/comments: 1) Equation 1 is an energy function, but I cannot see why it is a MRF. It seems that energy is computed considering only one single candidate 3D object proposal at a time. 2) Many assumption might be violated. For instance, if depths are sparse and with many errors, will the performance decrease significantly. If ground plane estimate failed, will the performance decrease significantly. I will suggest the authors to conduct some experiments to show how sensitive are these assumptions.
Summary: A 3D object bounding box proposal method is proposed to detect car, bike, pedestrian in stereo images. Since the focus is for automatic driving, the following strong assumption is used: 1) clean depth can be obtained from stereo pairs, 2) objects are on the same ground plane, 3) objects have typical 3D height and aspect ration which can be represented by a few template. According to these assumptions, several 3D cues such as 3d points density, free space amount, height prior, etc. are calculated (1.2s) to generate 3D object proposals. The proposed method obtains much higher recall than competing 2D and 3D methods given 500 proposals. By combining state-of-the-art CNN with the proposed region proposal method, significant accuracy improvement for hard to detected objects is achieved.

Author Feedback
Author rebuttal: We thank the reviewers for their comments.

To R1 and R2 on other datasets: In this paper we are interested in autonomous driving and our approach exploits domain specific properties, e.g., objects have typical 3D sizes and lie on the ground. We tested our approach on KITTI, which is the main benchmark for autonomous driving and showed state-of-the-art results on 2 out of 3 categories.
Our approach could be used in indoor scenes by exploiting, for example, room layout to calculate the floor and walls that limit the possible locations of objects. This is a very interesting avenue of future work.

To R3 and R6: Our approach is an MRF as an object proposal is encoded with 6 random variables: location, angle, class and template (see line 143).

R1:

Our approach uses class dependent proposals as for example different classes have different prototypical sizes

[24] takes 25min for images half the size and is thus impractical for autonomous driving applications.

R2: We will include 3D evaluation.

R3:

We do not use pairwise potentials between objects due to the additional complexity that this would entail. The goal of our method is to quickly propose a small set of possible object locations, which are then scored by the more costly CNN.

The performance of our object proposals show significant improvement compared to previous state of the art methods in all categories. For the task of object detection, the AP for pedestrians is ~2% worse than [13], but is ~9% higher than the previous CNN-based method. We suspect that the difference in performance might be due to the fact that in KITTI pedestrians are noisily annotated, and this confuses the strong CNN classifiers more than weaker ones.

R4:

NIPS call for papers solicits papers in the area of machine vision.Thus our paper is well aligned with NIPS.

We use Structured SVMs (S-SVM) to take into account *all* possible locations (thus avoiding hard negative mining) as well as the task loss. We used 3D overlap as the task loss, which is a natural loss function to compare two cuboids.

The strength of our approach is in exploiting class-specific information to generate object proposals, e.g., size of the objects. That being said, we will include the experiment suggested by the reviewer in the final version.

We use the sum over heights in Eq. 4 as it is a more robust measure (against noise in the stereo estimation) than the max.

We do not assume that objects are well segmented. The constraints encode the fact that two objects cannot occupy the same physical space. Furthermore, if we detect an object it has to be at least partially visible.

We choose C by cross-validation.

We will correct the typos and add details about ground-plane estimation. Note that we will release our code upon publication. We estimated the ground-plane by classifying superpixels using a neural net, and fitting a plane to the estimated ground pixels using RANSAC. We used the following features on the superpixels as input to the NN: mean RGB values, average 2D and 3D position (computed via depth), pitch and roll angles relative to the camera of the plane fit to the superpixel, a flag as to whether the average 2D position was above the horizon line, and standard deviation of both the color values and 3D position.

t is a discrete random variable where each possible state encodes a template (i.e., different 3D size).

y is bold as it contains a set of random variables.

R6:

Our approach takes 1.2s in a single core. As it is trivially parallelizable, real-time can be easily achieved.

Our approach depends on good stereo estimation, but can tolerate large amounts of noise, e.g., depth is very sparse and noisy far from the camera. We deal with this by sampling more boxes at larger distances. We will include an analysis of performance wrt distance to the camera in the revised version. Ground-estimation can't completely fail in our scenario, as the calibration of the camera rig is known and thus we can always go with the ground-plane prior.